# p97-dependent retrotranslocation and proteolytic processing govern formation of active Nrf1 upon proteasome inhibition

Senthil K Radhakrishnan[1], Willem den Besten[1], Raymond J Deshaies[2]*

[1]Division of Biology and Biological Engineering, California Institute of Technology, Pasadena, United States; [2]Division of Biology and Biological Engineering, Howard Hughes Medical Institute, California Institute of Technology, Pasadena, United States

**Abstract** Proteasome inhibition elicits an evolutionarily conserved response wherein proteasome subunit mRNAs are upregulated, resulting in recovery (i.e., 'bounce-back') of proteasome activity. We previously demonstrated that the transcription factor Nrf1/NFE2L1 mediates this homeostatic response in mammalian cells. We show here that Nrf1 is initially translocated into the lumen of the ER, but is rapidly and efficiently retrotranslocated to the cytosolic side of the membrane in a manner that depends on p97/VCP. Normally, retrotranslocated Nrf1 is degraded promptly by the proteasome and active species do not accumulate. However, in cells with compromised proteasomes, retrotranslocated Nrf1 escapes degradation and is cleaved N-terminal to Leu-104 to yield a fragment that is no longer tethered to the ER membrane. Importantly, this cleavage event is essential for Nrf1-dependent activation of proteasome gene expression upon proteasome inhibition. Our data uncover an unexpected role for p97 in activation of a transcription factor by relocalizing it from the ER lumen to the cytosol.

## Introduction

Transcription factor Nuclear factor erythroid derived 2-related factor 1 (Nrf1) belongs to the cap 'n' collar basic leucine zipper (CNC-bZIP) family of proteins that are known to be activated in response to cellular stress (*Sykiotis and Bohmann, 2010*). Other members of this family include p45 NF-E2, Nrf2, Nrf3, Bach1, and Bach2 (*Andrews et al., 1993*; *Moi et al., 1994*; *Oyake et al., 1996*; *Kobayashi et al., 1999*). The CNC-bZIP transcriptions factors heterodimerize with small Maf proteins (Maf F, Maf G, and Maf K) and preferentially bind to anti-oxidant response elements (AREs) present in the promoter region of their target genes (*Motohashi et al., 2002*). The most studied member of the CNC-bZIP family is Nrf2, which in response to oxidative stress directs a transcriptional program that helps maintain cellular redox homeostasis (*Kensler et al., 2007*). Owing to their sequence similarity, Nrf1 was initially thought to have an overlapping function with Nrf2 in regulating antioxidant gene expression, but a growing body of emerging evidence contradicts this notion.

Recently, we demonstrated that in mouse embryonic fibroblasts, Nrf1 but not Nrf2 is necessary for elevated expression of proteasome subunit mRNAs observed in cells treated with proteasome inhibitor, leading to a recovery or 'bounce-back' of proteasome activity (*Radhakrishnan et al., 2010*). Consistent with our observation, TCF11 (a longer isoform of Nrf1 found only in humans) was subsequently reported to be a mediator of proteasome bounce-back response after proteasome inhibition in human cells (*Steffen et al., 2010*). Thus it appears that Nrf1 functions to combat proteotoxic stress caused by proteasome inhibition in mammals akin to transcription factors RPN4 in yeast (*Xie and Varshavsky, 2001*) and Cnc-C in *Drosophila* (*Grimberg et al., 2011*). From the standpoint of cancer treatment, blockade of this bounce-back response may be a viable strategy to enhance the efficacy of proteasome inhibition therapy, given that Nrf1 depletion slows the rate of recovery of proteasome activity

*For correspondence: deshaies@caltech.edu

**eLife digest** Cells exposed to high temperatures, infections and other forms of stress often produce oxygen ions and peroxide molecules that can cause damage to proteins and DNA. Cells therefore rely on molecular machines called proteasomes to eliminate damaged proteins, before they cause too much harm. Two related transcription factors—proteins that interact with DNA to 'switch on' the expression of genes—are involved in a cell's responses to stress, but in different ways. Nrf2 switches on genes that limit the damage caused by oxygen ions and peroxide molecules, while Nrf1 switches on the genes that encode the components of the proteasome. As such, Nrf1 helps to restart proteasome activity if it has been shut off—a phenomenon known as 'bounce-back'.

Within a cell, Nrf1 is known to start off embedded within the membranes of a structure called the endoplasmic reticulum. However, it is not clear how activated Nrf1 leaves this membrane and enters the nucleus to interact with the cell's DNA. Now, Radhakrishnan et al. show that when Nrf1 is produced, most of its length is found inside the endoplasmic reticulum, with only a small piece being anchored in the surrounding membrane. This is unlike previously described transcription factors that associate with the endoplasmic reticulum, which are stuck to the outside of this structure.

Radhakrishnan et al. also discovered that the activation of Nrf1 depends on an enzyme called p97 or VCP. This enzyme helps to flip Nrf1 from the inside of the endoplasmic reticulum to its outside surface. In most cells, the proteasome then breaks down this part of Nrf1. However, if the proteasome is inhibited, an unknown enzyme cuts Nrf1 free from the endoplasmic reticulum, allowing it to migrate to the nucleus and promote the production of more proteasome components to counteract the inhibition.

Interestingly, drugs that inhibit the proteasome are used to combat cancer because the build-up of damaged proteins is toxic to the cancer cells. By showing that p97 promotes the 'bounce-back' of the proteasome, the work of Radhakrishnan et al. suggests that combining existing proteasome inhibitors with drugs that inhibit p97 could eventually lead to new, more effective, therapies for cancer or other diseases.

following transient application of a covalent proteasome inhibitor and results in enhanced proteasome inhibitor-mediated apoptosis in cancer cells (*Radhakrishnan et al., 2010*). However, since the molecular requirements for Nrf1 activation have not been described, there remains no known mechanism to exploit this possibility.

In addition to its role in the induced synthesis of proteasome subunit (PSM) genes, Nrf1 has also been found to regulate their basal expression in certain scenarios. For instance, in Nrf1$^{-/-}$ mouse neurons, PSM gene expression is diminished resulting in impaired proteasome function and neurodegeneration (*Lee et al., 2011*). Also, a liver-specific knockout of Nrf1 in mice caused a similar attenuation of PSM expression in hepatocytes (*Lee et al., 2013*).

Despite the importance of Nrf1 in proteasome biology and its potential as an anti-cancer target, a thorough understanding of the molecular mechanism behind its activation is currently lacking. What is known is that Nrf1 exists in two forms, p120 and p110, both of which are unstable and accumulate when the proteasome is inhibited (*Radhakrishnan et al., 2010*). Whereas p120 is embedded in the ER membrane, p110 is soluble and can enter the nucleus (*Biswas and Chan, 2009*). However, the mechanism leading to the formation of these species is controversial, with one study pointing to proteolytic cleavage (*Wang and Chan, 2006*) and another to differential glycosylation (*Zhang et al., 2007*). We show here that, in striking contrast to other ER membrane-tethered transcription factors, the bulk of the Nrf1 polypeptide is inserted into the ER lumen. Activation of Nrf1 depends on p97/VCP-dependent transfer of the luminal segments of Nrf1 to the cytosolic side of the ER membrane, followed by a novel proteolytic processing step.

## Results

### Transcriptionally active Nrf1 p110 is derived from p120 by proteolytic processing

To discriminate between the possibilities that p110 is derived by cleavage or deglycosylation of p120, we first overexpressed $^{3\times Flag}$Nrf1$^{HA}$ in human HEK-293T and mouse NIH-3T3 cells. This construct

contains three consecutive copies of the Flag tag at the N-terminus of Nrf1 and an HA tag at the C-terminus. In the presence of MG132, regardless of the cell type used, we observed that although the anti-HA antibody was able to detect two different forms of Nrf1 (~120 and ~110 kDa), the anti-Flag antibody was able to detect only the ~120 kDa form (*Figure 1A*). The simplest explanation for this observation is that Nrf1 p120 was cleaved somewhere close to the N-terminus to yield p110. To test this hypothesis and to identify the cleavage site, we overexpressed Nrf1[3×Flag] (Nrf1 with C-terminal triple Flag tag) in HEK-293T cells and immunopurified the protein and subjected the p120 and p110 forms to Edman degradation-based N-terminal sequencing. Although the ~120 kDa band confirmed the intact N-terminus of the full-length Nrf1, sequence from the ~110 kDa band was consistent with a new N-terminus starting with Leu-104 (*Figure 1B*; putative cleavage site indicated with a scissor).

To further confirm this observation, we introduced several mutations flanking the predicted cleavage site (m1 through m6 in *Figure 1B*) and compared these mutants to wild-type Nrf1[3×Flag] for their ability to be processed into Nrf1 p110. We found that all mutants were defective to varying degrees in their ability to be processed to the p110 form, although the defects of m4 and m6 were relatively modest (*Figure 1C*). Interestingly, m4 and m6 are the only mutants that retain Trp-103 at the P1 position (with respect to the cleavage site), implying a crucial role of this Trp residue for protease recognition.

Although our data suggested that p110 was derived from p120, there is no decisive evidence that proves that this is the case. To determine if there is a post-translational precursor–product relationship between Nrf1 p120 and p110 forms, we performed a pulse-chase experiment. We used L-azidohomoalanine (analog of the amino acid L-methionine) to pulse-label newly synthesized proteins and followed the resultant pool of Nrf1 during a chase period in which protein synthesis and proteasome-dependent degradation were both inhibited by the addition of cycloheximide and MG132. We observed that over time, wild-type p120 was processed to p110 but the non-cleavable Nrf1(m1)[3×Flag] was not processed (*Figure 1D*). Taken together, our data are consistent with a model in which newly synthesized Nrf1 p120 undergoes proteolytic processing between Trp-103 and Leu-104 to generate the p110 form.

Next, we asked if this proteolytic processing of Nrf1 is biologically relevant. To this end, we compared the ability of wild-type and non-cleavable mutant Nrf1 to reconstitute the bounce-back response in Nrf1[−/−] mouse embryonic fibroblasts (*Figure 2A*). As expected, wild-type Nrf1[3×Flag] was able to reinstate the induction of proteasome subunit (PSM) genes in response to MG132 treatment in these cells (*Figure 2B*). In contrast, non-cleavable Nrf1(m1)[3×Flag] was defective in inducing PSM genes under the same conditions, implicating a necessary role for proteolytic processing in Nrf1's transcriptional program.

## p97 is required for Nrf1 activity and its proteolytic processing

p97 is a homo-hexameric AAA ATPase that is implicated in numerous cellular processes ranging from cell-cycle regulation to membrane fusion and protein degradation (*Ye, 2006*). p97 has also been implicated in the turnover of the TCF11 isoform of human Nrf1 via the endoplasmic reticulum-associated degradation (ERAD) pathway (*Steffen et al., 2010*). Similar to inhibition of the proteasome, depletion of p97 leads to the strong accumulation of ubiquitin-conjugated substrates (*Wojcik et al., 2004*), underscoring its role in enabling degradation of a subset of the ubiquitinated proteome by the proteasome (*Kolawa et al., 2013*). Given that cells depleted of p97 or treated with proteasome inhibitors accumulate both ubiquitinated proteins and Nrf1, we asked if p97 depletion was also sufficient to trigger the Nrf1-mediated bounce-back response. To this end, we made use of NIH-3T3 mouse fibroblasts that have been engineered to stably express a doxycycline-inducible form of an shRNA targeting p97. Unlike proteasome inhibition, depletion of p97 upon addition of doxycycline did not induce the bounce-back response (*Figure 3A*; compare bars 1 and 3 for the PSM genes). By contrast, we observed that MG132-mediated bounce-back response was severely impaired when p97 was knocked-down (*Figure 3A*; compare bars 3 and 4 for the PSM genes).

To investigate the molecular basis for the lack of bounce-back response in p97-depleted cells, we examined the fate of Nrf1. Consistent with *Steffen et al., 2010*, Nrf1 accumulated in cells depleted of p97 (*Figure 3B*, compare lanes 1 and 3). To address the key question of whether p97 contributes to processing of Nrf1, which is essential for its activation, we evaluated the forms of Nrf1 that accumulated in control vs p97-depleted cells treated with MG132. Whereas the control cells accumulated p110 (indicative of Nrf1 activation), this form was markedly absent in p97-depleted cells (*Figure 3B*, compare lanes 2 and 4). This failure to process p120 is sufficient to account for the lack of bounce-back response in these cells. However, one concern was that due to the 3–4 days required for p97

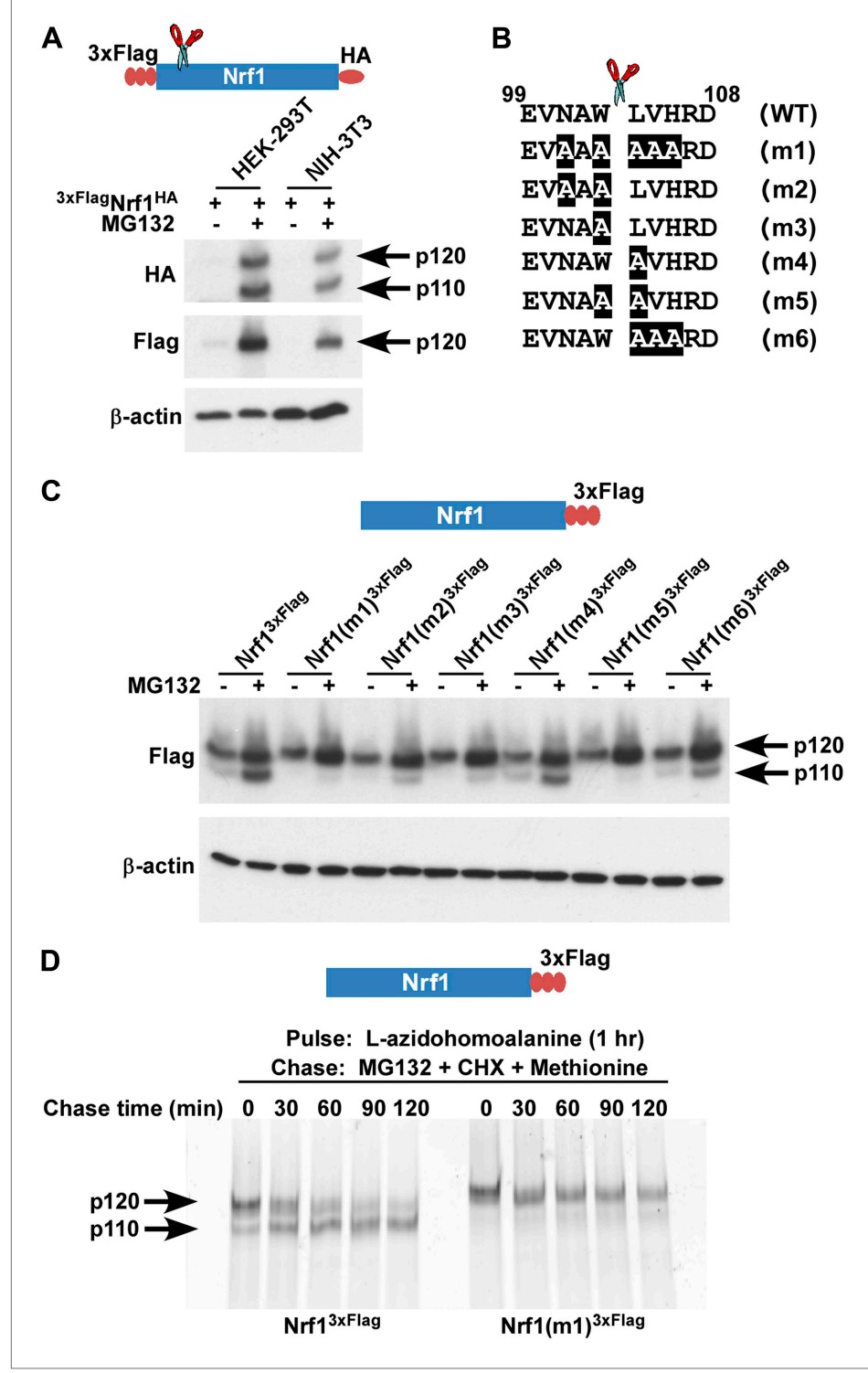

**Figure 1**. Nrf1 p110 is derived from p120 by proteolytic processing. (**A**) Human HEK-293T or mouse NIH-3T3 cells were transduced with a retrovirus expressing ³ˣFlagNrf1ᴴᴬ and 72 hr later were left untreated or treated with 5 μM MG132 for 5 hr. The cell lysates were then used for immunoblotting with anti-Flag and anti-HA antibodies. β-actin protein levels were evaluated as a loading control. (**B**) Amino acids 99 to 108 from human Nrf1 are shown. The scissor indicates the cleavage position predicted by the Edman degradation sequencing. Also shown are mutants 1 through 6 with the mutated amino acids highlighted. (**C**) HEK-293T cells were transiently transfected with wild-type or various mutant constructs (m1 through m6) of Nrf1³ˣFlag and 48 hr later were left untreated or treated

*Figure 1. Continued on next page*

*Figure 1. Continued*

with 5 µM MG132 for 5 hr. The cell lysates were then processed for immunoblotting with anti-Flag antibody. β-actin was used as a loading control. (**D**) HEK-293 cells stably expressing either wild-type Nrf1[3×Flag] or Nrf1(m1)[3×Flag] were pulse-labeled for 1 hr with L-azidohomoalanine and then chased with MG132, cycloheximide (CHX), and excess methionine. The cells were harvested at the time points indicated (from 0 min to 120 min) and the lysates were subjected to immunoprecipitation with anti-Flag beads. Nrf1 species were visualized using the tetramethylrhodamine based click-chemistry method as described in 'Materials and methods'.

knock-down, the p120 form we observed in p97-depleted cells could represent an aberrant dead-end species that only accumulated upon sustained deprivation of p97 activity, and was not a true intermediate in the processing pathway. To address this possibility, we made use of the recently described compound NMS-873, a reversible inhibitor of p97 (*Magnaghi et al., 2013*; *Polucci et al., 2013*). Treatment of NIH-3T3 mouse fibroblasts or HEK-293-Nrf1[3×Flag] cells with NMS-873 led to a robust accumulation of p120 Nrf1 (*Figure 3C*; lane 1), which was then efficiently converted to the p110 form with a half-life of ~30 min upon washout of NMS-873 in the presence of cycloheximide and MG132 (*Figure 3C*, lanes 2–6). The overproduced protein behaved similarly in HEK-293 cells, although the conversion of p120 to p110 was not as efficient as for the endogenous protein (*Figure 3C*, compare left and right panels). Thus, our results are consistent with a model in which p97 controls not only the degradation of Nrf1, but also its processing to the active p110 form.

## p97 re-positions the C-terminal domain of Nrf1 into the cytosol

It has been proposed that during synthesis, Nrf1 becomes anchored in the ER membrane via an N-terminal transmembrane region (residues 7–24) (*Zhang and Hayes, 2010*). Given the role of p97 in Nrf1 processing and activation, we hypothesized that either p97 promotes processing of cytosolically-exposed Nrf1 and possibly its subsequent extraction from binding partners (as is the case for Spt23 and Mga2 in yeast, *Hoppe et al., 2000*; *Rape et al., 2001*), or p97 could be involved in re-positioning the TAD and DBD of Nrf1 from the luminal to the cytosolic side of the ER membrane to facilitate its processing. To distinguish between these possibilities, we performed a series of protease protection experiments to assess the topology of Nrf1 under various conditions (*Figure 4A–D*). Regardless of p97 status (normal in the absence of doxycycline, or depleted in the presence of doxycycline, *Figure 4A*), [3×Flag]Nrf1 was susceptible to Proteinase K (PK) digestion in the absence of detergent, implying that the N-terminus was oriented towards the cytosol at all times. Interestingly, however, the orientation of the C-terminus of Nrf1[3×Flag] showed a strong dependence on p97 status (*Figure 4B*). In mock-depleted cells, Nrf1[3×Flag] was largely susceptible to digestion by PK (*Figure 4B*, compare lanes 4 and 5). In stark contrast, when p97 was depleted by the addition of doxycycline, Nrf1[3×Flag] was largely resistant to PK (*Figure 4B*, compare lanes 7 and 8, or lanes 10 and 11). The different behaviors of [3×Flag]Nrf1 and Nrf1[3×Flag] and the failure to observe detectable trimming of Nrf1[3×Flag] are consistent with a topology wherein only a very small amino-terminal segment of newly-synthesized Nrf1 is exposed to the cytosol (amino acids 1–6), whereas the bulk of Nrf1 (amino acids 25–742) remains in the lumen of the ER when the retrotranslocation activity of p97 is blocked. Interestingly, when we overexposed our blots, a small, protease-resistant pool of p120 was detectable at steady-state (*Figure 4C*, lanes 1 and 2). This pool quickly disappeared upon chasing with cycloheximide plus MG132 (lanes 4–11). We propose that retrotranslocation of p120 constitutively occurs at such a fast rate that, at steady-state, only a tiny fraction of luminal p120 can be detected unless retrotranslocation is blocked by depletion of p97 as in *Figure 4B*.

If the type II transmembrane orientation ($N_{cytosol}/C_{lumen}$) for Nrf1 indicated by the protease protection experiments represents a true intermediate in its biogenesis, the accumulation of this species should be reversed upon restoration of p97 activity. To test if this is the case, we utilized the p97 inhibitor NMS-873. Whereas Nrf1[3×Flag] was largely resistant to digestion by PK in HEK293 cells treated with NMS-873 (*Figure 4D*, lanes 1 and 2), it became sensitive to PK upon washout of NMS-873 in the presence of MG132 plus cycloheximide (*Figure 4D*, lanes 4 and 5). In this case, the Nrf1 was detected as a p120 species, because the tagged protein was chased to p110 less efficiently than the endogenous protein (*Figure 3C*), and the p110 that did form during the chase fractionated with soluble proteins upon preparation of the membrane fraction. Nrf1(m1)[3×Flag] mirrored the behavior of the wild-type protein (*Figure 4D*), indicating that the lack of processing observed for non-cleavable mutant was not due to defective retrotranslocation.

**Figure 2**. Non-cleavable Nrf1 mutant is functionally defective. (**A**) Nrf1$^{-/-}$ MEFs were transfected with constructs expressing either Nrf1$^{3\times Flag}$, Nrf1(m1)$^{3\times Flag}$, or vector control and 48 hr later were subjected to Geneticin selection. These antibiotic resistant cells were treated with 1 μM MG132 for 10 hr as indicated. Lysates from these cells were immunoblotted with anti-Flag antibodies to detect exogenous Nrf1. β-actin was used as a loading control. (**B**) RNA from Nrf1$^{-/-}$ MEFs transfected and selected as described above was used for quantitative RT-PCR to assess the mRNA levels of representative PSM genes. The values were normalized to GAPDH mRNA levels. Error bars denote SD (n = 3).


To corroborate the results from the protease protection experiments, we also monitored the glycosylation status of Nrf1. We reasoned that when the C-terminal domain (CTD) is oriented towards the lumen, Nrf1 should be glycosylated on a set of Asn-X-Ser/Thr sites C-terminal to the transmembrane domain (***Zhang et al., 2007***), whereas when the CTD is retrotranslocated to the cytosol, it should become degylcosylated. To evaluate the glycosylation status of Nrf1, we monitored its sensitivity to the degylcosylating enzyme Endoglycosidase H (Endo H). Nrf1 p120 that accumulated in cells treated with NMS-873 was sensitive to Endo H (***Figure 4E***, compare lanes 1 and 2 or 5 and 6), whereas it

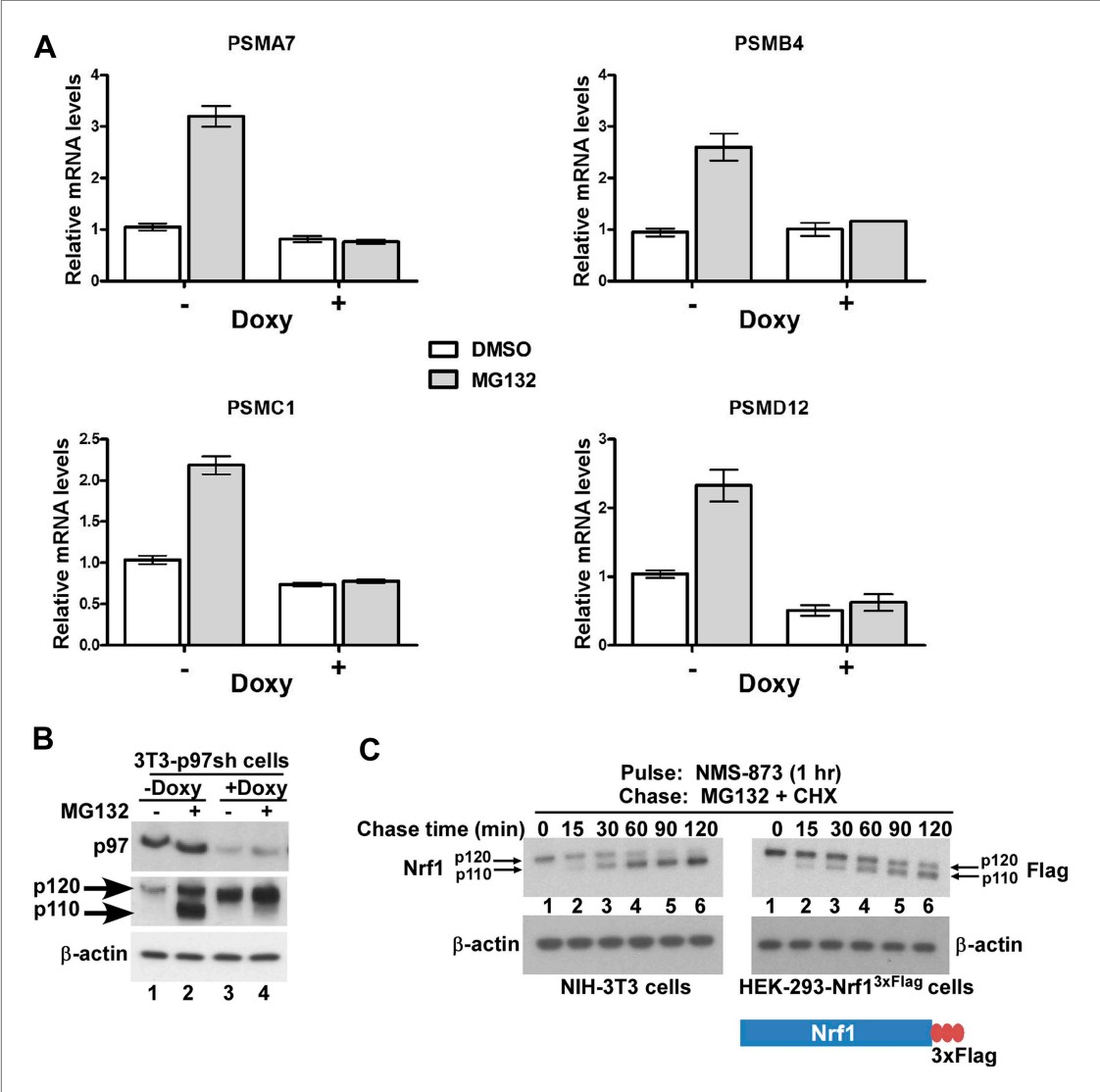

**Figure 3**. p97 is required for processing and activation of Nrf1. (**A**) NIH-3T3-p97sh cells stably expressing doxycycline (Doxy)-inducible shRNA targeting p97 were either mock-treated or induced with Doxy (1 µg/ml) for 3 days after which the cells were further treated with DMSO or 1 µM MG132 as indicated for 10 hr. The RNA from these cells was used for quantitative RT-PCR to assess the mRNA levels of the representative PSM genes. The values were normalized to GAPDH mRNA levels. Error bars denote SD (n = 3). (**B**) NIH-3T3-p97sh cells were subjected to Doxy and MG132 treatments as above and the cell lysates were fractionated by SDS-PAGE and immunoblotted to detect p97 and Nrf1. β-actin was used as a loading control. (**C**) NIH-3T3 (left panel) or HEK-293-Nrf1[3xFlag] cells (right panel) were pulsed for an hour with 10 µM NMS-873 and then chased with MG132 plus cycloheximide (CHX). The cells were harvested at the time points indicated (from 0 min to 120 min) and the lysates were fractionated by SDS-PAGE and immunoblotted to detect endogenous Nrf1. β-actin was used as a loading control.

became largely resistant to Endo H upon being chased in the presence of MG132 plus cycloheximide (*Figure 4E*, compare lanes 3 and 4 or 7 and 8). This behavior is consistent with the protease protection data that indicated that the CTD of p120 flips from a luminal to a cytosolic orientation upon restoration of p97 activity. Notably, the cytosolically-oriented Nrf1 that accumulated upon restoration of p97 activity migrated more slowly than expected for Nrf1 that has been deglycosylated and proteolytically processed. This was particularly evident when the migration of the primary translation product for amino acids 104–742 was compared to the species of Nrf1 that accumulated in cells subjected to various perturbations (*Figure 4—figure supplement 1*). Our data suggest that once it is retrotranslocated to the cytosol, Nrf1 is not only deglycosylated and proteolytically cleaved after Trp103, but it also gains one or more additional modifications, the nature of which remains unknown.

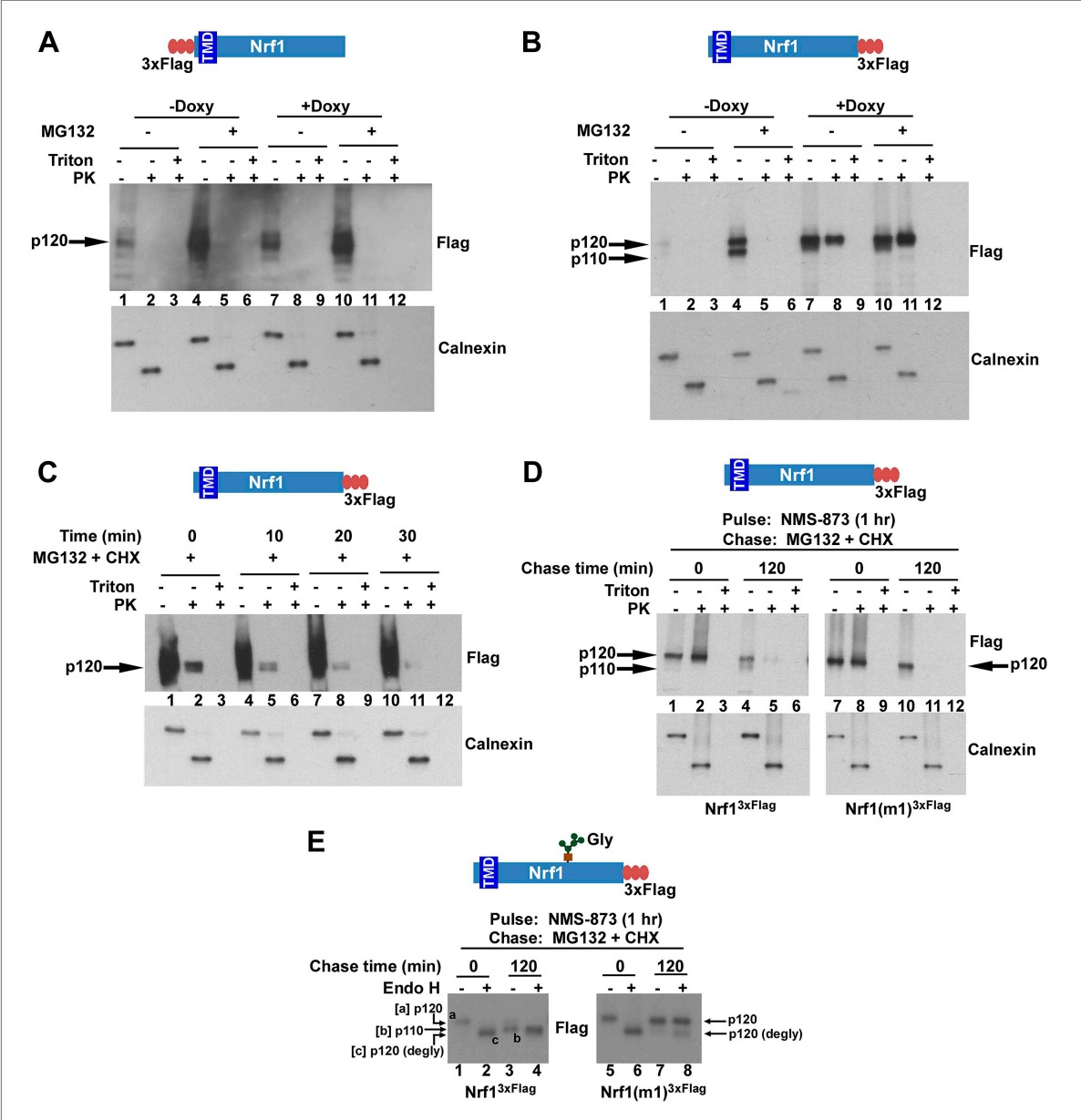

**Figure 4**. The C-terminus of Nrf1 is re-positioned from the lumen to the cytosol in a p97-dependent manner. (**A**) HEK-293 cells stably expressing doxycycline (Doxy)-inducible shRNA targeting p97 (HEK-293-p97sh cells) were transiently transfected with a [3xFlag]Nrf1 expression construct and 24 hr later were incubated in the presence or absence of Doxy for 4 days after which the cells were further treated or not with 5 µM MG132 for 5 hr. Microsomes prepared from these cells were subjected to protease protection assay with Proteinase K (PK). SDS-PAGE followed by immunoblotting with anti-Flag antibody was used to detect Nrf1. Formation of a discrete cleavage product of Calnexin in the absence of detergent (Triton X-100) was used as an indicator for intact microsomes. (**B**) Same as (**A**), except that cells were transfected with an Nrf1[3xFlag] expression construct. (**C**) HEK-293 cells stably expressing the non-cleavable Nrf1(m1)[3xFlag] were treated with MG132 plus cycloheximide (CHX) and cells were harvested at indicated time points. Microsomes from these cells were subjected to protease protection assay with PK. Samples were fractionated by SDS-PAGE followed by immunoblotting with anti-Flag antibody. (**D**) HEK-293 cells stably expressing either Nrf1[3xFlag] or the non-cleavable Nrf1(m1)[3xFlag] were pulsed for an hour with 10 µM NMS-873 and then chased with MG132 plus cycloheximide (CHX). Microsomes prepared from cells harvested at the indicated time points (0 min and 120 min) were used in a protease protection assay with PK. SDS-PAGE followed by immunoblotting with anti-Flag antibody was used to detect Nrf1. (**E**) Total cell lysates from the pulse-chase experiment described above were subjected to deglycosylation by the enzyme Endoglycosidase H prior to SDS-PAGE and immunoblotting with anti-Flag. Species 'a' refers to full-length Nrf1 (p120) that was fully glycosylated. Species 'b' refers to Nrf1 that was proteolytically processed to p110 and deglycosylated following p97-dependent retrotranslocation. Species 'c' refers to species 'a' that was deglycosylated with Endo H. Note that species 'b' migrates slightly more slowly than species 'c' even though both species were deglycosylated and species 'b' lacked the

*Figure 4. Continued on next page*

*Figure 4. Continued*

N-terminal 103 residues. The unexpectedly slow mobility of species 'b' presumably reflects acquisition of additional post-translational modifications upon retrotranslocation on Nrf1's CTD to the cytosolic side of the ER membrane. (TMD–transmembrane domain; Gly–N-linked glycan).

The following figure supplements are available for figure 4:

**Figure supplement 1**. Different forms of Nrf1.

## Discussion

Mouse Nrf1 and its human ortholog TCF11 are transcription factors that enhance the accumulation of mRNAs that encode proteasome subunits upon inhibition of proteasome activity (*Radhakrishnan et al., 2010*; *Steffen et al., 2010*). Prior work established that Nrf1 is initially integrated into the ER membrane but is turned over rapidly (*Biswas and Chan, 2009*; *Radhakrishnan et al., 2010*). Upon proteasome inhibition, it accumulates, migrates to the nucleus, and activates gene expression. In this work, we make several critical new observations about the biogenesis and processing of Nrf1 that allow us to formulate a model that explains how it becomes activated in response to proteasome inhibition. Specifically, we show that the domains of Nrf1 that are implicated in transcriptional activation are initially translocated into the lumen of the ER, but subsequently are retrotranslocated to the cytosolic side of the membrane in a manner that depends on the AAA ATPase p97/VCP. In unperturbed cells, the retrotranslocated Nrf1 is promptly degraded by the proteasome, resulting in a futile cycle of continuous synthesis, membrane insertion, retrotranslocation, and degradation (*Figure 5*, left-hand side). However, in cells with compromised proteasome activity, some of the retrotranslocated Nrf1 escapes degradation and is cleaved on the N-terminal side of Leu-104 to yield a p110 fragment that is no longer tethered to the ER membrane and can migrate to the nucleus (*Figure 5*, right-hand side). Importantly, this cleavage event is essential for Nrf1-dependent activation of proteasome gene expression upon inhibition of the proteasome.

This model raises two important questions. First, what is the identity of the protease that clips p120 to yield p110? This remains unclear, but the available evidence rules out the most likely candidates. Cleavage by the site-1 and site-2 proteases involved in mobilization of SREBP appears unlikely, because CHO cell lines deficient in these activities (*Rawson et al., 1997*, *1998*) were nonetheless able to accumulate p110 upon proteasome inhibition (R Rawson, personal communication). In addition, the proteasome, which cleaves Spt23 and Mga2 from the ER membrane in yeast (*Hoppe et al., 2000*), appears to be dispensable for cleavage because we observed robust formation of p110 in two different cell lines with three different proteasome inhibitors across a wide range of concentrations (*Figure 1*, and data not shown). Finally, an inhibitor of ER-localized rhomboid protease(s) (*Pierrat et al., 2011*) failed to impede the accumulation of p110 in cells treated with MG132 (data not shown). The second question is why does p110 not normally accumulate in unperturbed cells? Although it will be easier to address this question when the protease is in hand, a reasonable working hypothesis is that in unperturbed cells, p97-dependent retrotranslocation and proteasome-dependent degradation are so intimately coupled that there is no opportunity for the cleavage event to occur as Nrf1 is threaded from the membrane into the maw of the proteasome. However, when the proteasome is inhibited, Nrf1 accumulates on the cytosolic side of the membrane where it is susceptible to cleavage. Even if the protease occasionally intercedes before the proteasome can act and small amounts of p110 are generated, p110 is intrinsically unstable (*Radhakrishnan et al., 2010*) and thus does not accumulate to substantial levels.

It should be immediately evident from the foregoing discussion how the activation of Nrf1 differs markedly from the activation of all other ER membrane-tethered transcription factors that have been studied to date, including SREBP, ATF6, OASIS, CREB-H, Spt23, and Mga2 (*Brown and Goldstein, 1997*; *Hoppe et al., 2000*; *Ye et al., 2000*; *Kondo et al., 2005*; *Zhang et al., 2006*). In all of those cases, the precursor is inserted into the ER membrane such that the functional domains involved in transcription are displayed on the cytosolic side of the membrane. Cleavages in the cytosolic or intramembrane domains can therefore liberate soluble fragments that translocate to the nucleus. By contrast, since Nrf1 begins its life with its functional domains in the ER lumen, it must be retrotranslocated to the cytosol before processing can yield a functional fragment that can relocalize to the nucleus. However, because the p97-dependent retrotranslocation pathway traversed by Nrf1 is normally employed to clear misfolded proteins from the ER (*Wolf and Stolz, 2012*), the inevitable consequence of Nrf1 piggybacking on this mechanism is that it is rapidly degraded by the proteasome as soon as it

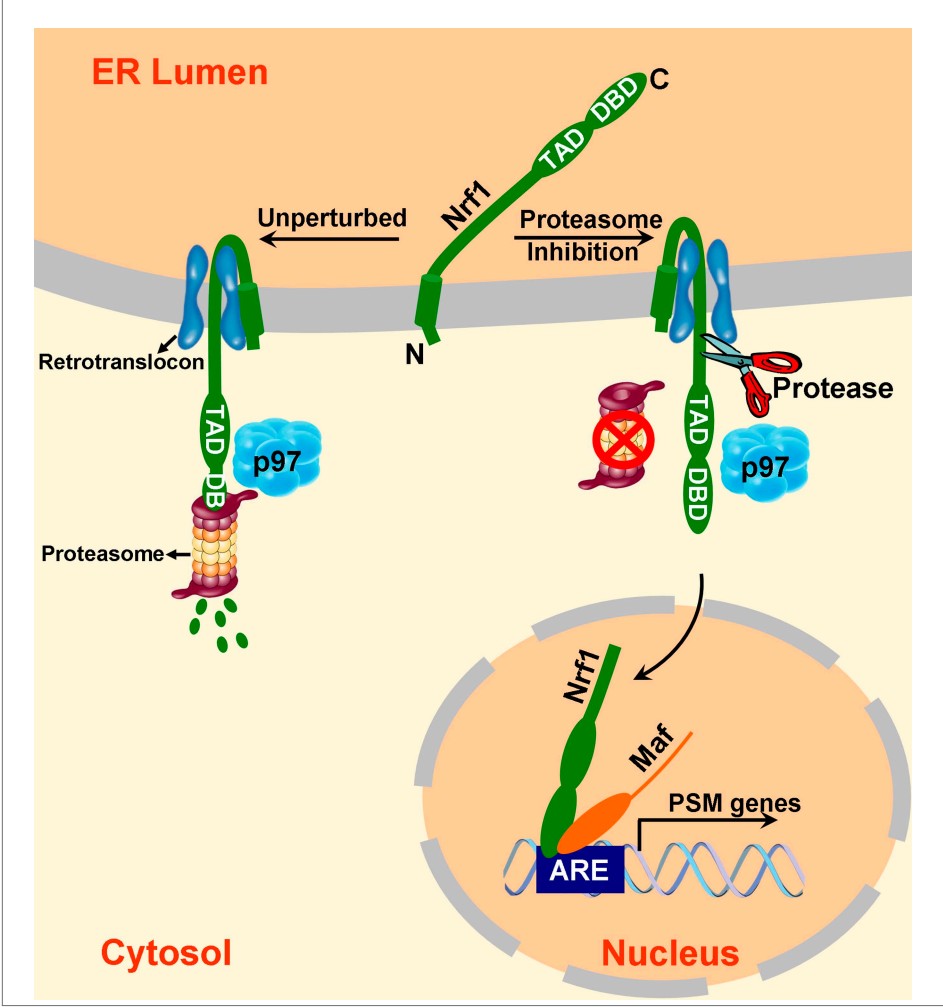

**Figure 5**. A model to explain Nrf1 activation. Newly synthesized Nrf1 p120 is initially inserted into the endoplasmic reticulum (ER) membrane in an $N_{cytosol}/C_{lumen}$ orientation. It is then rapidly and efficiently retrotranslocated to the cytosol, where it is immediately degraded by the proteasome. In cells deficient in proteasome activity, degradation of cytosolically-exposed Nrf1 is retarded, allowing sufficient time for proteolytic processing of Nrf1 p120 to yield the active p110 form which can then migrate to the nucleus, heterodimerize with small Maf proteins, and activate transcription of its targets (e.g., PSM genes) by binding to antioxidant response elements (ARE). Note that the exact mechanism of the retrotranslocation step is unknown; what is shown in the figure reflects one possibility among several. Also, although the putative protease is depicted to be in the cytosol, it remains possible that it is in the ER membrane or lumen.

gains access to the cytosol. This precludes the accumulation of Nrf1 precursor on the cytosolic side of the ER membrane, except under circumstances where there is insufficient proteasome activity to keep pace with the appearance of substrates.

Recently, Zhang and Hayes (*Zhang and Hayes, 2013*) put forth a model for Nrf1 activation that shares some features with the model proposed here but is very different in its key aspects, leading to a fundamentally different view regarding the biogenesis and maturation of p120 into an active transcription factor. Specifically, Zhang and Hayes proposed that p120 spans the ER membrane at least three times. They argued that turnover of this species is promoted by calpain, not the proteasome. p120 that escapes degradation was envisioned to remain stably integrated as a multispanning trans-membrane protein until it is mobilized by an unknown signal, which triggers retrotranslocation of the luminal domains followed by their deglycosylation to yield active Nrf1. It was proposed that Nrf1 is later cleaved by the proteasome to yield lower molecular weight soluble species, but appearance of active Nrf1 precedes this cleavage event. However, the causal relationship between cleavage and

activation could not be evaluated because the cleavage site(s) was not identified. Another key distinction is that Zhang and Hayes do not perform true chase experiments, and thus their data do not distinguish whether there is a precursor–product relationship between the various Nrf1 species, or they arise from newly-synthesized Nrf1 following different biogenetic pathways. We do not understand the basis for the multiple discrepancies between our data and those of Zhang and Hayes.

Our observation that a functional transcription factor can begin its life in the lumen of the ER is unexpected but not without precedent. Ricin, a plant toxin, and some bacterial toxins including cholera toxin, and shiga toxin likewise are retrotranslocated from the ER lumen to the cytosol, where they exert their biological functions (*Inoue et al., 2011*). Another example is the Hepatitis E Virus capsid protein ORF2, which co-opts the ERAD machinery to gain access to the cytosol (*Surjit et al., 2007*). To our knowledge, the only host protein that undergoes retrotranslocation from the ER but escapes degradation by the proteasome is calreticulin (*Afshar et al., 2005*; *Gold et al., 2010*). Upon inhibition of the proteasome, presumably the entire complement of ERAD substrates mimics Nrf1 in that these proteins are retrotranslocated from the luminal to the cytosolic side of the ER membrane but are not degraded. Our observations provoke the question of whether some of these proteins, like Nrf1, exert novel functions upon their accumulation in the cytosol. An obvious question that emerges from our findings is why is Nrf1 regulated by such a baroque mechanism? Targeting Nrf1 to the ER may ensure that the basal level of Nrf1 activity is essentially zero in the absence of proteasome stress, due to the normally tight coupling between retrotranslocation and degradation (*Wolf and Stolz, 2012*). In cells that experience insufficient proteasome activity—brought about either through environmental insults including proteasome or chaperone inhibitors produced by microbes, or through changes in gene expression—accumulation of active Nrf1 can restore an adequate level of proteasome function.

Our finding that p97 is required for the Nrf1-mediated proteasome bounce-back response potentially has therapeutic implications. Previously, we demonstrated that in cancer cells, depletion of Nrf1 slows down recovery of proteasome activity upon transient inhibition of the proteasome and enhances apoptosis caused by a covalent proteasome inhibitor (*Radhakrishnan et al., 2010*). Although inhibition of transcription factors with small molecules has proven difficult (*Berg, 2008*), p97 is a druggable target (*Chou et al., 2011*, *2013*; *Magnaghi et al., 2013*). Indeed, recent papers have reported enhanced killing of cancer cells when proteasome and p97 inhibitors are used in combination (*Auner et al., 2013*; *Chou et al., 2013*). It will be interesting to see whether blockade of the Nrf1-mediated bounce-back response through inhibition of p97 enhances the efficacy of proteasome inhibitor therapy in multiple myeloma or enables expansion of proteasome inhibitor therapy into new indications.

## Materials and methods

### Constructs

$^{3\times Flag}$Nrf1 (RDB-2411) and $^{3\times Flag}$Nrf1$^{HA}$ (RDB-2412) constructs have been described previously (*Radhakrishnan et al., 2010*). Human Nrf1 coding region along with a C-terminal 3×Flag sequence was cloned into pcDNA3.1+ (Invitrogen, Carlsbad, CA) to establish the wild-type construct Nrf1$^{3\times Flag}$ (RDB-2867). The following mutants were derived from Nrf1$^{3\times Flag}$ via site-directed mutagenesis using the indicated forward primers together with their corresponding reverse complement primers: Nrf1(m1)$^{3\times Flag}$ (RDB-2868; forward primer: 5′-ACA GGT TCC AGG TGC AAC CA CTG AGG TAG CTG CCG CGG CGG CTG CCC GAG ACC CAG AGG G-3′), Nrf1(m2)$^{3\times Flag}$ (RDB-2869; forward primer: 5′-GGT GCC AAC CAC TGA GGT AGC TGC CGC GCT GGT TCA CCG AGA-3′), Nrf1(m3)$^{3\times Flag}$ (RDB-2870; forward primer: 5′-CAC TGA GGT AAA TGC CGC GCT GGT TCA CCG AGA C-3′), Nrf1(m4)$^{3\times Flag}$ (RDB-2871; forward primer: 5′-GAG GTA AAT GCC TGG GCG GTT CAC CGA GAC C-3′), Nrf1(m5)$^{3\times Flag}$ (RDB-2872; forward primer: 5′-CCA CTG AGG TAA ATG CCG CGG CGG TTC ACC GAG ACC CAG-3′), and Nrf1(m6)$^{3\times Flag}$ (RDB-2873; forward primer: 5′-ACT GAG GTA AAT GCC TGG GCG GCT GCC CGA GAC CCA GAG GGG TC-3′).

The doxycycline-inducible shRNA expression construct pTRIPZ-p97sh (RDB-2874) targeting p97 was based on a 21-mer sequence (AAC AGC CAT TCT CAA ACA GAA) present in the coding region of both human and mouse genes and was cloned into the pTRIPZ vector (Open Biosystems, Huntsville, AL).

### Cell culture

HEK-293T, HEK-293, and NIH-3T3 cell lines were grown in Dulbecco's modified Eagle's medium (DMEM) supplemented with 10% fetal bovine serum (Atlanta Biologicals, Norcross, GA), penicillin, and streptomycin (Invitrogen) at 37°C in 5% $CO_2$. Immortalized Nrf1$^{-/-}$ mouse embryonic fibroblasts

(MEFs) (*Radhakrishnan et al., 2010*) were grown as above except that the medium was additionally supplemented with β-mercaptoethanol and non-essential amino acids (Invitrogen).

## Transient transfection and retroviral transduction

Transfection reagents Lipofectamine 2000 (for HEK-293T, HEK-293 cells) or Lipofectamine LTX (for mouse embryonic fibroblasts) were used for transient transfections as per manufacturer's recommendations (Invitrogen).

For retroviral production, HEK-293T cells were transfected with the required retroviral construct along with helper plasmids. 48 hr after transfection, media supernatant containing the retrovirus was collected every 4–5 hr for 2 days. This retrovirus-containing medium, supplemented with polybrene (10 µg/ml), was used to transduce the target cells.

## Stable cell lines

NIH-3T3-p97sh cell line (DTC-138) expressing doxycyline-inducible p97-specific shRNA was generated by transducing the mouse NIH-3T3 cells with the pTRIPZ-p97sh retroviral construct and selecting them in the presence of 7.5 µg/ml of puromycin. The HEK-293-p97sh cell line (DTC-139) was generated similarly except that 2 µg/ml puromycin was used for selection.

Stable cell lines HEK-293-Nrf1$^{3×Flag}$ (DTC-140) and HEK-293-Nrf1(m1)$^{3×Flag}$ (DTC-141) were established by transfecting HEK-293 cells with the constructs Nrf1$^{3×Flag}$ and Nrf1(m1)$^{3×Flag}$ respectively and subjecting the cells to 500 µg/ml Geneticin selection.

## Immunoblot analysis

Cells were lysed in RIPA buffer (50 mM Tris pH 7.4, 150 mM NaCl, 1% NP40, 1% Na.Deoxycholate, 0.1% SDS, 1 mM EDTA) supplemented with a protease and phosphatase inhibitor cocktail (Pierce, Rockford, IL). Immunoblots were performed with antibodies specific for Nrf1 (8052S; Cell Signaling, Danvers, MA), Flag tag (A8592; Sigma–Aldrich, St. Louis, MO), HA tag (2013819; Roche Diagnostics, Indianapolis, IN), p97 (sc-20799; SantaCruz Biotechnology, Dallas, TX), Calnexin (sc-11397; SantaCruz Biotechnology), and β-actin (A5441; Sigma–Aldrich).

## N-terminal Edman degradation sequencing

HEK-293T cell line was transiently transfected with the construct Nrf1$^{3×Flag}$ and 48 hr later was treated with 5 µM MG132 for 5 hr. Cell lysate was prepared in lysis buffer (50 mM Tris pH 7.4, 0.5 M NaCl, 1 mM EDTA, 1% Triton X-100) supplemented with protease and phosphatase inhibitor cocktail (Pierce). The lysate was then subjected to immunoprecipitation with anti-Flag beads (Sigma-Aldrich) and eluted with Flag peptide (Sigma-Aldrich). The eluate was resolved by SDS-PAGE and transferred on to a PVDF membrane in the presence of 10 mM CAPS (3-cyclohexylamino-1-propane sulfonic acid), pH 11.0 and 10% Methanol. The membrane was stained with Ponceau S solution, and the Nrf1-specific bands (p120 and p110) were excised and sequenced using a protein micro-sequencer (Procise cLC 492A sequencer; Applied Biosystems, Foster City, CA).

Whereas the p120 sample confirmed the intact N-terminus, the p110 sample indicated a mix of amino acid sequences LVHRD and VHRD. The latter sequence is likely to be a frayed-end version of the former that was generated during sample processing (F Rusnak, personal communication), thus revealing the N-terminus of p110 to be Leu-104.

## Quantitative reverse transcription PCR

RNA was isolated using the RNeasy kit (Qiagen, Valencia, CA). cDNA was prepared using the Superscript III first strand synthesis kit (Invitrogen) according to the manufacturer's recommendations. Quantitative PCR (qPCR) was performed using the SYBR GreenER supermix (Invitrogen). The primers used to quantify murine proteasome subunits (PSM) and GAPDH mRNA levels have been described previously (*Radhakrishnan et al., 2010*).

## Pulse-chase assay

The pulse-chase experiment to establish the precursor–product relationship between Nrf1 p120 and p110 was carried out using the Click-iT Metabolic labeling and detection kits (Invitrogen) as per the manufacturer's recommendations. Briefly, HEK-293 cells stably expressing either wild-type Nrf1$^{3×Flag}$ or Nrf1(m1)$^{3×Flag}$ were starved for an hour in methionine-free medium and pulsed for an additional hour with 50 µM L-azidohomoalanine. Then the cells were washed with PBS and subjected to a chase

with the addition of 5 µM MG132, 50 µg/ml cycloheximide, and 2 mM L-methionine. The cells were harvested at different time points and the cell lysates were prepared in lysis buffer (50 mM Tris pH 7.4, 0.5 M NaCl, 1 mM EDTA, 1% Triton X-100) supplemented with protease and phosphatase inhibitor cocktail (Pierce). The lysates were then used for immunoprecipitation with anti-Flag beads (Sigma-Aldrich). The immunoprecipitates were labeled with Tetramethylrhodamine (TAMRA)-alkyne and resolved by SDS-PAGE. The labeled bands were visualized in a Typhoon laser scan imaging system (GE Healthcare, Pittsburgh, PA).

### Protease protection assay

The cells were washed once in PBS, resuspended in 10 mM Hepes-KOH pH 7.5 buffer and incubated on ice for 10 min. The swollen cells were then sedimented, resuspended in homogenization buffer (10 mM Hepes-KOH pH 7.5, 10 mM KCl, 1.5 mM $MgCl_2$, 5 mM EGTA, and 250 mM sucrose) and passed through a 27G syringe needle several times. The homogenate was then subjected to serial centrifugations at 600×$g$ (10 min), 3000×$g$ (10 min) and 100,000×$g$ (60 min). The microsomes collected at the end of the 100,000×$g$ ultracentrifugation step were resuspended in membrane buffer (10 mM Hepes-KOH pH 7.5, 50 mM KOAc, 2 mM Mg(OAc)$_2$, 1 mM DTT, and 250 mM sucrose).

Microsomes were mock-treated or subjected to Proteinase K (0.5 µg/µl) treatment either in the absence or presence of 1% Triton X-100 for 60 min on ice. The samples were then precipitated by trichloroacetic acid (TCA), resuspended in boiling sample buffer, resolved by SDS-PAGE and subsequently analyzed by immunoblotting with appropriate antibodies.

### Deglycosylation assay

Cells were lysed in IP buffer (50 mM HEPES/KOH pH 7.5, 5 mM Mg(OAc)$_2$, 70 mM KOAc, 0.2% Triton X-100; 10% glycerol, and 0.2 mM EDTA) supplemented with a protease and phosphatase inhibitor cocktail (Pierce). These cell lysates were used for deglycosylation reactions with the enzyme Endoglycosidase H for 1 hr at 37°C as per the manufacturer's recommendations (New England Biolabs, Ipswich, MA).

## Acknowledgements

We are grateful to the members of the Deshaies' lab for helpful advice and critical reading of the manuscript. We thank R Rawson (UT Southwestern Medical Center at Dallas) for testing Nrf1 processing in S1P/S2P deficient CHO cells. We gratefully acknowledge the help of J Zhou and F Rusnak (Protein/ Peptide Micro Analytical Laboratory, California Institute of Technology) for N-terminal sequencing of Nrf1. We thank Han-Jie Zhou and BioDuro, LLC for the synthesis of Rhomboid inhibitor. We thank Lev G Lis and Michael A Walters (University of Minnesota) for the synthesis of NMS-873. RJD is a Howard Hughes Medical Institute (HHMI) Investigator, and this work was supported in part by HHMI.

## Additional information

### Competing interests

RJD: RJD is a founder and shareholder of Cleave Biosciences, and an *eLife* reviewing editor. The other authors declare that no competing interests exist.

### Funding

| Funder | Grant reference number | Author |
| --- | --- | --- |
| Howard Hughes Medical Institute | | Raymond J Deshaies |
| National Cancer Institute's Howard Temin Pathway to Independence Award K99/R00 | K99CA154884 | Senthil K Radhakrishnan |

The funders had no role in study design, data collection and interpretation, or the decision to submit the work for publication.

### Author contributions

SKR, Conception and design, Acquisition of data, Analysis and interpretation of data, Drafting or revising the article; WB, Analysis and interpretation of data, Contributed unpublished essential data or reagents; RJD, Conception and design, Analysis and interpretation of data, Drafting or revising the article

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
