## [Decision Letter]

[Editors’ note: although it is not typical of the review process at *eLife*, in this case the editors decided to include the reviews in their entirety for the authors’ consideration as they prepared their revised submission.]

Thank you for sending your work entitled “p97-dependent retrotranslocation and proteolytic processing govern formation of active Nrf1 upon proteasome inhibition” for consideration at *eLife*. Your article has been favorably evaluated by a Senior editor and three reviewers, two of whom are members of our Board of Reviewing Editors.

The reviewers have made some suggestions that you should consider prior to submitting a final manuscript. The most important question is whether you wish to revise Figure 5 to indicate that the proposed model for C-terminal retrotranslocation led by the C-terminus is only provisional and other models are also possible. We are including the reviews below for your information.

*Reviewer #1*:

This important paper documents a novel mechanism for control of proteasome biosynthesis through the p57-mediated retrotranslocation of Nrf1 from ER lumen to cytosol and eventually to the nucleus. The experiments are comprehensive and well performed.

I see only one major problem. The diagram in Figure 5 implies that Nrf1 is retrotranslocated with the COOH-terminus leading the way. This requires a new membrane-spanning segment that the authors do not document. It seems equally likely that the NH2-terminal transmembrane helix is actually an uncleaved signal sequence that may never dissociate from the Sec61 translocation machinery. In this case the retrotranslocation may proceed by simply reversing the original membrane insertion mechanism so that the NH2- terminus would lead the way. If the authors agree they should consider modifying Figure 5 to indicate this alternate mechanism for retrotranslocation.

*Reviewer #2*:

This manuscript reports on a mechanism for the regulated activation of the transcription factor Nrf1 in mammalian cells with diminished proteasome activity. Using a panel of suitably tagged mutant versions of Nrf1 and exploiting inhibitors of the AAA protein p97 and the proteasome, Radhakrishnan and colleagues provide compelling evidence that the bulk of Nrf1 (its C-terminus) is normally translocated into the lumen of the ER and then subject to proteasome-dependent ER-associated degradation (a process i.e. dependent on the activity of p97). Under conditions of diminished proteasome activity, the C-terminal portion of Nrf1 is untethered from its membrane anchor by a yet-to-be identified protease (that cleaves after Trp103) and finds its way to the nucleus where it serves to activate the transcription of genes encoding proteasome subunits.

This study is remarkable both in terms of its physiological significance (clarifying a mechanism relevant to the regulated expression of a key component of the cell's protein degradation apparatus) and in terms of the level of detail it provides on the molecular aspects. The cleavage site of Nrf1 is precisely mapped and, crucially, the proposed unusual itinerary of Nrf1's C-terminal portion is confirmed by tracing the glycosylation status of the various trapped intermediates. Thus, this study will be of interest to a wide range of investigators.

Two specific comments follow (the first critical and the second speculative):

The critical comment:

The supplement to Figure 4 is a bit cryptic. The experiment exploits cotransin - presumably the compound described by Taunton's group in 2005 (PMID: 16015336) *–* to inhibit Sec61-mediated translocation of something. But neither the logic of the experiment nor the provenance of cotransin (Taunton's paper is not cited), nor the significance of the observations are discussed in any detail. *eLife*'s readers would benefit from rectification of these shortcomings.

The speculative comment:

Comparing the mobility of the P110 fragment synthesized in vitro, with that isolated from cells, the researchers conclude that the latter is subject to a novel post-translational modification. This is likely true, but it is worth perhaps considering the possibility that the modification in question may be a direct consequence of cytosolic deglycosylation of the single (predicted) N-linked glycan in Nrf1, which would convert Asn 347 to an aspartic acid (EndoH results in a different remnant, a GlcNAc attached to Asn 347 accounting perhaps for the different mobility of bands c and b in Figure 4). If the deglycosylated residue were important to P110's activity, this could provide a satisfying explanation for Nrf1's unusual itinerary: its purpose is to convert Asn347 to Asp347. This idea would be supported (though by no means proven) by judicious mutagenesis of Asn347.

*Reviewer #3*:

This manuscript demonstrates a novel transcriptional regulatory mechanism. The authors show that the transcriptional activation domain of Nrf1 is localized in the ER lumen after its initial synthesis, and then subject to constitutive retrotranslocation followed by rapid proteasomal degradation under normal circumstance. Upon inhibition of proteasomes, a proteolytic event-generated C-terminal fragment of the protein (p110) activates transcription of genes encoding proteasome subunits. These results are potentially interesting, but they are not enough to support the model shown in Figure 5, primarily because the authors fail to show how the precursor of Nrf1 (p120) associates with membranes.

1) The authors cited a previous study showing that Nrf1 (p120) is an intrinsic membrane protein. However, after reading the cited reference, this reviewer found that the particular study did not address whether the protein is really an intrinsic membrane protein or a soluble ER luminal protein. Indeed, the reviewer suspects that Nrf1 (p120) may be a soluble ER luminal protein as the so-called N-terminal transmembrane segment is not hydrophobic enough to be a transmembrane helix but appears to be a good candidate for a signal sequence. This hypothesis can be tested by alkaline treatment or other methods to break microsome membranes to see whether such treatment releases the protein from membrane pellets in cells in which p97 is depleted. An intrinsic ER membrane protein such as calnexin and a soluble ER luminal protein such as calreticulin should be used as controls in the experiment.

2) If NRF1 (p120) is a soluble ER protein, then after retrotranslocation it should be a soluble cytosolic protein. Under this condition, the proteolytic cleavage may allow p110 to enter the nucleus where it activates transcription of its target genes. This hypothesis should be tested by cell fractionation experiments to determine the subcellular localization of p120 and p110 in cells treated with a proteasomal inhibitor but with p97 present.

3) Even if NRF1 (p120) is an intrinsic membrane protein, the model shown in Figure 5 still does not make sense as it suggests the presence of a new transmembrane domain during the retrotranslocation. Is there any evidence supporting this hypothesis? A more likely model is that during retrotranslocation the transmembrane helix horizontally dips into membranes so that both the N and C terminal segment of the proteins can face the cytosol. If this is the case, the authors need to demonstrate that whereas p120 is an intrinsic membrane protein, p110 is a soluble protein that can enter the nucleus.

---

## [Author Response]

Reviewer #1:

*This important paper documents a novel mechanism for control of proteasome biosynthesis through the p57-mediated retrotranslocation of Nrf1 from ER lumen to cytosol and eventually to the nucleus. The experiments are comprehensive and well performed*.

*I see only one major problem. The diagram in*
Figure 5
*implies that Nrf1 is retrotranslocated with the COOH-terminus leading the way. This requires a new membrane-spanning segment that the authors do not document. It seems equally likely that the NH2-terminal transmembrane helix is actually an uncleaved signal sequence that may never dissociate from the Sec61 translocation machinery. In this case the retrotranslocation may proceed by simply reversing the original membrane insertion mechanism so that the NH2-terminus would lead the way. If the authors agree they should consider modifying*
Figure 5
*to indicate this alternate mechanism for retrotranslocation*.

We were attempting to imply as little as we possibly could because we do not really know in detail how the retrotranslocation process works. We originally drew the “new” transmembrane domain as being within the retrotranslocon, to imply that what is shown is an intermediate in the retrotranslocation/degradation process. However, we felt that this made the drawing excessively complicated. Another possibility in addition to the one mentioned by the reviewer is that the N-terminal membrane anchor domain switches polarity during the retrotranslocation process. Given that our efforts at simplification appear to have backfired, what we have done is to modify the drawing to include a putative retrotranslocon, and then state explicitly in the text of the figure legend that the drawing reflects one possibility among several.

Reviewer #2:

*[…] This study is remarkable both in terms of its physiological significance (clarifying a mechanism relevant to the regulated expression of a key component of the cell's protein degradation apparatus) and in terms of the level of detail it provides on the molecular aspects. The cleavage site of Nrf1 is precisely mapped and, crucially, the proposed unusual itinerary of Nrf1's C-terminal portion is confirmed by tracing the glycosylation status of the various trapped intermediates. Thus, this study will be of interest to a wide range of investigators*.

*Two specific comments follow (the first critical and the second speculative)*:

*The critical comment*:

*The supplement to*
Figure 4
*is a bit cryptic. The experiment exploits cotransin - presumably the compound described by Taunton's group in 2005 (PMID: 16015336) – to inhibit Sec61-mediated translocation of something. But neither the logic of the experiment nor the provenance of cotransin (Taunton's paper is not cited), nor the significance of the observations are discussed in any detail. eLife's readers would benefit from rectification of these shortcomings*.

*The speculative comment*:

*Comparing the mobility of the P110 fragment synthesized in vitro, with that isolated from cells, the researchers conclude that the latter is subject to a novel post-translational modification. This is likely true, but it is worth perhaps considering the possibility that the modification in question may be a direct consequence of cytosolic deglycosylation of the single (predicted) N-linked glycan in Nrf1, which would convert Asn 347 to an aspartic acid (EndoH results in a different remnant, a GlcNAc attached to Asn 347 accounting perhaps for the different mobility of bands c and b in*
Figure 4*). If the deglycosylated residue were important to P110's activity, this could provide a satisfying explanation for Nrf1's unusual itinerary: its purpose is to convert Asn347 to Asp347. This idea would be supported (though by no means proven) by judicious mutagenesis of Asn347*.

Nrf1 actually contains 9 potential glycosylation sites but it is not known how many of them are modified. We presume from the context of his or her comment that the reviewer is referring to the bands b and e in Figure 4—figure supplement 1, not bands b and c. Although it is possible that the difference in mobility between these bands is due to conversion of a set of Asn residues to Asp, this seems unlikely given that the shift in MW is on the order of ∼10 kD. Nevertheless, we now mention this possibility in the revised figure legend. It is possible that conversion of Asn to Asp results in activation of Nrf1, and in fact it has been suggested that this is the case (Zhang et al., 2009 Biochem J. 418, 293-310). In the Zhang study, they created a mutant in which 7 putative glycosylation sites were changed to aspartate. They reported enhanced activity for the mutant construct on a synthetic reporter, but only two-fold (Figure 3 of their paper). But, there are a number of problems with their experiment. First of all, it was done by transient transfection and there is no information regarding the degree of overexpression. Second, the authors did not quantify their Western blot of the wild type and mutant constructs and it actually appears that there might be a bit more of the mutant protein, so it is unclear if increased transcription of the reporter gene is due to increased expression or increased specific activity. Finally, it was not established that each of the sites that is mutated is actually glycosylated. Even if the Zhang result is correct, a two-fold effect is of questionable significance. This experiment would need to be done under more physiological conditions to test properly the interesting hypothesis raised by the reviewer.

Reviewer #3:

*This manuscript demonstrates a novel transcriptional regulatory mechanism. The authors show that the transcriptional activation domain of Nrf1 is localized in the ER lumen after its initial synthesis, and then subject to constitutive retrotranslocation followed by rapid proteasomal degradation under normal circumstance. Upon inhibition of proteasomes, a proteolytic event-generated C-terminal fragment of the protein (p110) activates transcription of genes encoding proteasome subunits. These results are potentially interesting, but they are not enough to support the model shown in*
Figure 5*, primarily because the authors fail to show how the precursor of Nrf1 (p120) associates with membranes*.

*1) The authors cited a previous study showing that Nrf1 (p120) is an intrinsic membrane protein. However, after reading the cited reference, this reviewer found that the particular study did not address whether the protein is really an intrinsic membrane protein or a soluble ER luminal protein. Indeed, the reviewer suspects that Nrf1 (p120) may be a soluble ER luminal protein as the so-called N-terminal transmembrane segment is not hydrophobic enough to be a transmembrane helix but appears to be a good candidate for a signal sequence. This hypothesis can be tested by alkaline treatment or other methods to break microsome membranes to see whether such treatment releases the protein from membrane pellets in cells in which p97 is depleted. An intrinsic ER membrane protein such as calnexin and a soluble ER luminal protein such as calreticulin should be used as controls in the experiment*.

The reviewer’s argument is contradicted by the data we show in Figure 4. In cells depleted of p97, essentially 100% of the accumulated p120 molecules have an N-terminal Flag tag removed in a protease protection experiment conducted in the absence of detergent (panel **A**), whereas a C-terminal Flag tag is completely protected (panel **B**). The most logical conclusion from this experiment is that the N-terminus faces the cytosol and the C-terminus faces the lumen (i.e., Nrf1 cannot be completely luminal). Regarding the point about the N-terminal transmembrane domain, we analyzed the Nrf1 sequence using the transmembrane domain prediction program TMpred (http://www.ch.embnet.org/software/TMPRED_form.html), which indicated a strong transmembrane domain (amino acids 7 to 26).

For the sake of thoroughness, we have also done membrane extraction experiments to address this issue, but the results have not been straightforward. About half of the Nrf1 is carbonate extractable in cells depleted of p97 or treated with p97 inhibitor (by contrast, 100% of calnexin resists carbonate extraction in both cases). By contrast most of Nrf1 is carbonate extractable in cells treated with MG132, consistent with the idea that it has been dislocated from the ER membrane. In cells treated with both p97 and proteasome inhibitors, the p97-deficient phenotype is epistatic. Thus, while the protease protection suggests that essentially all Nrf1 molecules at a p97 block are transmembrane, the carbonate extraction suggests that roughly 50% of these transmembrane species remain in an aqueous compartment. The simplest resolution of this is to hypothesize that a significant fraction of full-length Nrf1 molecules remain associated with the translocon, reminiscent of what was recently suggested for Deg1-Sec62 chimeras (Rubenstein et al., 2012 J. Cell Biol. 197:761-773). Nailing this point down would require considerable more experimentation but would not alter our fundamental conclusion that the C-terminus of Nrf1 starts out its life in the ER lumen and p97 is required for its retrotranslocation, processing, and activity. Hence we decided to not include the extraction data.

*2) If NRF1 (p120) is a soluble ER protein, then after retrotranslocation it should be a soluble cytosolic protein. Under this condition, the proteolytic cleavage may allow p110 to enter the nucleus where it activates transcription of its target genes. This hypothesis should be tested by cell fractionation experiments to determine the subcellular localization of p120 and p110 in cells treated with a proteasomal inhibitor but with p97 present*.

It has been previously established that p110 is nuclear (e.g., [32]. J Biol Chem, 281:19,676-19,687), so we did not feel it was necessary to repeat these observations. As part of the extraction experiments described in the response to point #1 we observed that p110 largely sediments but is readily extractable from the pellet with either carbonate or non-ionic detergent, consistent with the hypothesis that it is no longer anchored by a transmembrane domain.

*3) Even if NRF1 (p120) is an intrinsic membrane protein, the model shown in*
Figure 5
*still does not make sense as it suggests the presence of a new transmembrane domain during the retrotranslocation. Is there any evidence supporting this hypothesis? A more likely model is that during retrotranslocation the transmembrane helix horizontally dips into membranes so that both the N and C terminal segment of the proteins can face the cytosol. If this is the case, the authors need to demonstrate that whereas p120 is an intrinsic membrane protein, p110 is a soluble protein that can enter the nucleus*.

What the reviewer proposes is very similar to our suggestion, described in the response to Reviewer #1, that the transmembrane domain may switch polarity during retrotranslocation. As described in more detail in the response to Reviewer #1, we have redrawn the figure and modified the legend to make more clear what we originally intended to show.